# Mediterranean diet assessment challenges: Validation of the Croatian Version of the 14-item Mediterranean Diet Serving Score (MDSS) Questionnaire

**Mario Marendić**[1,2], **Nikolina Polić**[3], **Helena Matek**[4], **Lucija Oršulić**[5], **Ozren Polašek**[6], **Ivana Kolčić**[6]*

1 University Department of Health Studies, University of Split, Split, Croatia, 2 University Postgraduate Doctoral Study Program Evidence-Based Medicine, University of Split School of Medicine, Split, Croatia, 3 Institute of Emergency Medicine of Split-Dalmatia County, Split, Croatia, 4 General Hospital Šibenik, Šibenik, Croatia, 5 University of Split School of Medicine, Split, Croatia, 6 Department of Public Health, University of Split School of Medicine, Split, Croatia

* ikolcic@mefst.hr

**Data Availability Statement:** The data underlying the results presented in the study are available

## Abstract

Mediterranean diet (MD) is among the most commonly investigated diets and recognized as one of the healthiest dietary patterns. Due to its complexity, geographical and cultural variations, it also represents a challenge for quantification. The aim of this cross-sectional study was to assess reliability and validity of the Croatian version of the 14-item Mediterranean Diet Serving Score (MDSS), using the Mediterranean Diet Adherence Screener (MEDAS) as the referent test. We included the exploratory sample of 360 medical students, and a confirmatory sample of 299 health studies students from the University of Split, Croatia. Test-retest reliability and validity of the MDSS were tested using intra-class correlation coefficients (ICC), while Cohen's kappa statistic was used to test correct classification of subjects into MD adherent/non-adherent category. A very good reliability was shown for the overall MDSS score (ICC = 0.881 [95% CI 0.843–0.909]), and a moderate reliability for the binary adherence (κ = 0.584). Concurrent validity of the MDSS was also better when expressed as a total score (ICC = 0.544 [0.439–0.629]) as opposed to the adherence (κ = 0.223), with similar result in the confirmatory sample (ICC = 0.510 [0.384–0.610]; κ = 0.216). Disappointingly, only 13.6% of medical students were adherent to the MD according to MDSS, and 19.7% according to the MEDAS questionnaire. Nevertheless, MDSS score was positively correlated with age (ρ = 0.179: P = 0.003), self-assessed health perception (ρ = 0.123; P = 0.047), and mental well-being (ρ = 0.139: P = 0.022). MDSS questionnaire is a short, reliable and reasonably valid instrument, and thus useful for assessing the MD adherence, with better results when used as a numeric score, even in the population with low MD adherence.

from: https://figshare.com/search?q=10.6084%2Fm9.figshare.13560497.

**Funding:** The author(s) received no specific funding for this work.

**Competing interests:** The authors have declared that no competing interests exist.

## Introduction

Nutrition has a profound impact on health, both in the short-term and life-long scale. Nutrition affects the disease burden of both infectious and non-communicable disease outcomes. According to the Global Burden of Disease Study for 2017, dietary risk factors are accountable for as many as 11 million deaths and 255 million DALYs worldwide, with the biggest contribution from high sodium intake, low intake of whole grains and fruit [1].

Unhealthy diet is contributing to both poor (or insufficient) nourishment and environmental degradation, which points to the urgent need for a global transformation of the food system, ideally back towards traditional diets [2]. An example of such a diet is the Mediterranean diet (MD) [2, 3]. The MD is one of the most commonly investigated dietary patterns, with many beneficial effects for human health described so far [4], playing a role in prevention of cardiovascular diseases (CVD) [5–7], CVD risk factors [8], diabetes [9, 10], cancer [11, 12], protection of mental health [13, 14], and better health-related quality of life [15].

According to the revised MD pyramid, the guidelines for adults include high daily intake of vegetables, fruits, whole grains, and olive oil, moderate daily consumption of nuts, dairy products, and red wine, weekly intake of legumes, fish, eggs and poultry, and overall low intake of red and processed meat and other processed foods [16]. The MD is an incredibly rich nutritional pattern, with many varieties of dishes, flavors, textures, and nutrients, creating a complex "exposome", which is the reason why it represents a challenge for defining and measuring [17].

In general, measuring nutrition and eating habits is far from being simple. There are several different approaches and all of them have their advantages and limitations. Dietary pattern can be defined using a general description, dietary pyramids, *a priori* scoring systems, *a posteriori* dietary pattern formation, or by quantifying food and nutrient content [18, 19]. *A priori* approach is more commonly applied and it uses a predefined scoring system, i.e. a diet index, in case of the MD an "attempt to make a global evaluation of the quality of the diet based on a traditional Mediterranean reference pattern" [20]. The indexes are usually based on data acquired within a 24 h quantitative intake recall, dietary records or food frequency questionnaires (FFQs) [21]. FFQ is one of the most commonly used approach for dietary assessment, usually showing good reproducibility and validity for MD assessment in heterogeneous samples [22]. Unfortunately, FFQs include an exhaustive number of questions that take a long time to answer [23]. To overcome this issue, numerous short indexes have been developed to assess the adherence to the overall MD pattern. Because they are useful in rapidly evaluating a patient's eating habits, such brief questionnaires are of special interest to researchers in the field of nutrition, as well as public health professionals and clinicians. Ideally, dietary questionnaires should not require much time to complete, should be easily and quickly evaluated, and interpretation of the result should not require elaborate knowledge on nutrition [24]. Mediterranean Diet Adherence Screener (MEDAS) [25], and the Mediterranean Diet Serving Score (MDSS) are examples of such indexes [26].

MEDAS emerged within the PREDIMED study (*Prevención con Dieta Mediterránea*), one of the most comprehensive experimental studies in nutrition to date, aimed at investigating the long-term effects of the MD on incident CVD in high risk individuals [27]. This questionnaire transcended its original use in the Spanish population, and has been used widely in various cultural and societal settings. Several validation studies showed that MEDAS is a valid and reliable questionnaire in different countries and languages [28–33].

MDSS is another example of a simple and short scoring approach, and it incorporates 14 food groups in exact accordance with the new MD pyramid [16, 26], which is an important advantage of this index. According to the original study, MDSS is a valid instrument [26].

Despite existence of many dietary indexes used in literature for assessing compliance with the MD, the evidence on their applicability and psychometric quality is scarce [34]. Due to the differences in the design of the studies, as well as the reliability and validity of the instruments, it is not possible to determine which questionnaire is the best.

Even though Croatia is one of the seven Mediterranean countries that participated in the process of inscription of the MD to the UNESCO's representative list of the intangible cultural heritage of humanity, no study has so far tested any dietary questionnaire regarding its validity. Only one previous study has assessed the reliability of the KIDMED questionnaire in a sample of students from the continental part of Croatia, and not from the Mediterranean region [35]. This study aims to evaluate validity (accuracy) and test-retest reliability of the Croatian version of the short, 14-item Mediterranean Diet Serving Score (MDSS) questionnaire, compared to the Mediterranean Diet Adherence Screener (MEDAS), based on a sample of students from the University of Split, Croatia.

## Materials and methods

### Study design and subjects

This cross-sectional study was carried out in Split, Croatia, the largest city on the coast of the Adriatic Sea. We included two independent samples in order to assess psychometric properties of the Croatian version of a short MD questionnaire. The initial, exploratory sample of medical students from the University of Split School of Medicine was used for reliability testing (test-retest repeatability) and concurrent and construct validity of the Croatian version of the short MDSS questionnaire. A total of 377 medical students enrolled in the first, third and fifth study year (out of six study years) were sampled during the period of December 2018—October 2019, with the overall response rate of 80.2%. The second confirmatory sample was used to replicate results and confirm the initial MDSS questionnaire validity results and to investigate its' predictive validity. This independent confirmatory sample consisted of 320 students from the University Department of Health Studies (nurses, lab technicians, radiology technicians, physiotherapists; response rate 81.2%) sampled during the period of May—December 2019. Inclusion criteria were age over 18 years, both genders, and the willingness to provide informed consent. There were no exclusion criteria.

### Procedures

The MEDAS questionnaire was selected as the reference (gold standard) to validate the MDSS questionnaire in Croatian language, due to its broad use in the literature and previous results in several validation studies [28–33]. The first step was to translate both the MDSS and MEDAS questionnaires into Croatian, using the originally proposed and validated instruments [23, 26]. This was done according to the guidelines from the International Society for Pharmacoeconomics and Outcomes Research (ISPOR, Fig 1) [36].

In short, an independent, proficient English speaker did a back-translation (from Croatian to English), followed by a second back translation (from English to Croatian). A certified English translator compared the original and back-translated versions of the English questionnaire, while a professor of the Croatian language compared two translated Croatian versions. All discrepancies were resolved, and Croatian versions of both questionnaires were harmonized. Finally, we performed a pilot study including 51 students from the University Department of Health Studies to test applicability of the questionnaires. Namely, we aimed to obtain information about respondent and administrative burden, and to assess the cultural acceptability and language applicability [34], these responses were not used for validity or reliability

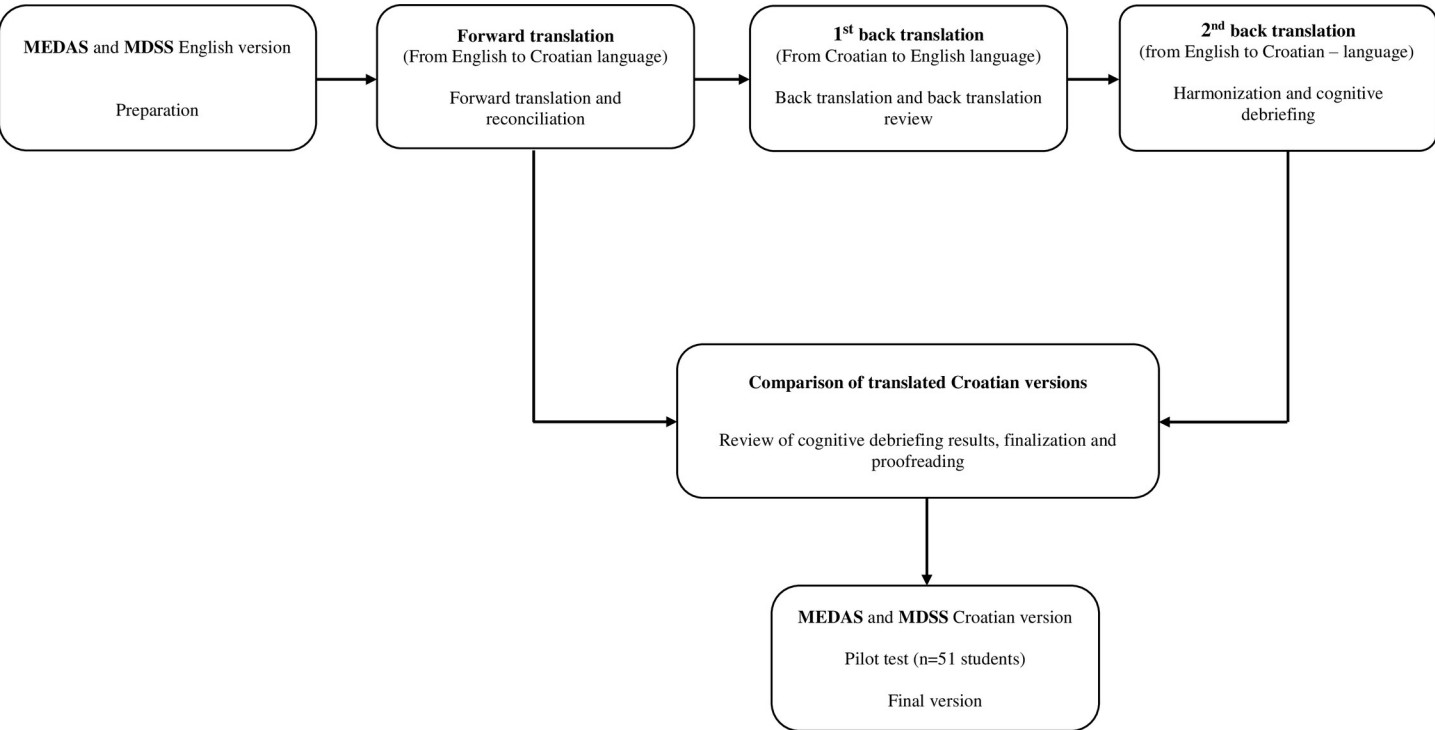

**Fig 1. Flow diagram of translation method, according to the guidelines of ISPOR [36].** The same translation procedure was used for both the Mediterranean Diet Serving Score (MDSS) questionnaire [26] and the Mediterranean Diet Adherence Screener (MEDAS) questionnaire [23].

analysis. There were no major objections by the involved students, and the questionnaires were finalized accordingly. Detailed flow chart of the study is presented in Fig 2.

The students were invited to participate in the study during their mandatory courses, in order to ensure the highest possible response rate. After the initial explanation of the purpose and procedures of the study, students who decided to participate were asked to provide informed consent. Medical students filled out the questionnaire twice, whereas health studies (nursing) students just once, the latter group filled out a more detailed survey. Since students answered the questionnaire anonymously, they were asked to provide a unique code using letters and numbers related to their identity (the first letter of their parent's name, the first letter of the place they were born, and two starting digits of their birthday date). This code was needed to pair the data obtained during the first and second time point (test and retest), simultaneously avoiding unnecessary memorization of the codes [37]. The retest was carried out between one and two weeks after the first round of data collection (Fig 2), which is acceptable for evaluating test-retest reliability [38]. All of the data were collected using a paper-based, self-administered approach, and the average time needed to complete the survey was 10 minutes for medical students and 15 minutes for health studies students. During this time, a facilitator was present (at least one of the authors of this study during each surveying sessions), ready to assist with any questions regarding the survey posed by respondents.

## Questionnaire

Subjects in the exploratory sample completed an anonymous self-administered questionnaire consisting of general questions (gender, age, study program, and the year of study) and two short MD questionnaires.

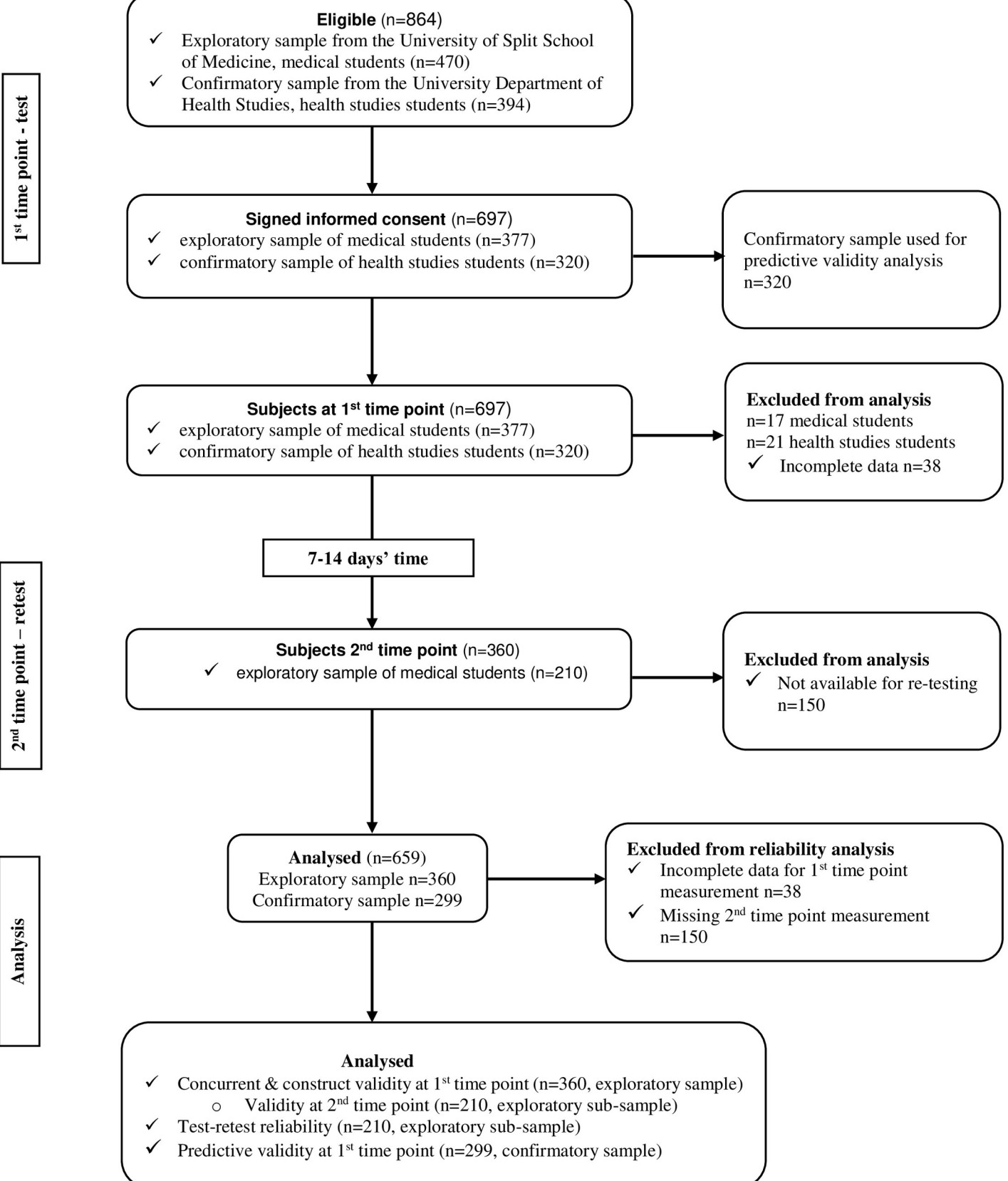

**Fig 2. Flow diagram of the validation study.** All stages of data collection are depicted, along with the sample size included in each stage and the final analysis of the data.

The confirmatory sample of subjects was additionally asked about other characteristics and habits needed for predictive validity assessment, such as body weight, body height, how long ago they had weighed themselves (in days), and smoking habits (active smokers, ex-smokers, non-smokers). Using body weight and height, we calculated the body mass index using the standard formula:

$$\text{BMI} = \frac{weight\ (kg)}{height\ (m^2)}$$

Additionally, health studies students answered questions on eating habits, including the number of meals per day (both main meals and snacks, separately for working days and week-end days), how often they usually eat breakfast (days per week), and whether they cook (possible answers were "yes, frequently", "sometimes" or "no"). We also asked whether they have ever been on a weight loss diet ("yes" or "no"), whether they are satisfied with their body appearance ("yes", "no" or "I don't think about it"), and whether they are snacking while watching TV (possible answers "yes, frequently", "yes, sometimes" or "no"). Questions on physical activity included practicing any sports ("yes, weekly", "rarely" or "never") and going to the gym ("yes, weekly", "rarely" or "never"). Finally, we asked students to rate their self-perceived health and quality of life using a Likert scale, where 0 represented a very poor health or quality of life, and 10 represented full health or quality of life.

Additionally, we used the Warwick–Edinburgh Mental Well-being Scale (WEMWBS), a validated questionnaire used for measuring mental well-being, especially focusing on the positive aspects of mental health [39]. This questionnaire was translated to Croatian using the same ISPOR procedure [36], and it was applied in both exploratory and confirmatory sample. The purpose of WEMWBS questionnaire was to serve as a non-dietary reliability comparator and as an outcome in the predictive validity analysis in order to investigate the association between the MD and well-being in students.

**MD assessment instruments.** We used two short MD questionnaires in this study. The MDSS questionnaire [26] was being validated and the MEDAS questionnaire [23] was selected as the gold standard, due to its extensive previous use in the literature.

The Mediterranean Diet Serving Score (MDSS) is an MD index that was originally validated against the Mediterranean Dietary Score (MDS), proposed by Trichopoulou et al. [40]. In the validation study on 1,155 women aged 12–83 years from Spain, both MDSS and MDS indexes were based on the data obtained from the semi-quantitative food frequency questionnaire (FFQ) [26]. It was found that MDSS index was "an updated, easy, valid, and accurate instrument to assess MD adherence based on the consumption of foods and food groups per meal, day, and week" [26], while being in accordance with the latest update of the Mediterranean Diet Pyramid [16, 26]. MDSS index incorporates 14 typical MD food groups, and individuals whose intake is within the recommended range receive either 3, 2, or 1 points for each of the specific food groups consumption per meal, daily or weekly, while those individuals who don't reach the particular goal get 0 points (Table 1).

Therefore, the MDSS index can range between 0 and 24 points for adults and between 0 and 23 for adolescents, since alcoholic beverages intake is not considered appropriate in this age group [26]. Out of the maximum 24 points, 12 points (50%) can be obtained for recommended intake of fruits, vegetables, cereals, and olive oil (3 points each, for consumption during every main meal). Additional 4 points can be obtained for daily intake of dairy products

**Table 1. Comparison of the MEDAS and MDSS questionnaires and the scoring procedure (each row represents one question that contributes to the overall score).**

| Items | MEDAS [23] | MEDAS scoring (points) | MDSS [26] | MDSS scoring (points) |
|---|---|---|---|---|
| Cereals | *Not included* | / | 1–2 servings/main meal | 3 |
| Olive oil as the principal source of fat for cooking | Yes | 1 | *Not included* | / |
| Olive oil frequency per day | ≥4 Tbsp | 1 | 1 serving/main meal | 3 |
| Vegetables per day | ≥2 (≥1 portion raw or as a salad) | 1 | ≥2 servings/main meal | 3 |
| Fruits per day | ≥3 (including natural fruit juices) | 1 | 1–2 servings/main meal (not including fruit juices) | 3 |
| Dairy products per day | *Not included* | / | 2 servings | 2 |
| Nuts per day/week | ≥3 servings per week | 1 | 1–2 servings per day | 2 |
| Legumes per week | ≥3 servings | 1 | ≥2 servings | 1 |
| Fish/shellfish per week | ≥3 servings | 1 | ≥2 servings | 1 |
| Chicken | preferentially consume chicken, turkey, or rabbit meat instead of veal, pork, hamburger, or sausage | 1 | 2 servings per week | 1 |
| Red/processed meat | <1 serving per day of red meat, hamburger, or meat products like ham, sausage, etc. | 1 | 2 servings per week (only red meat, not including processed meat) | 1 |
| Eggs per week | *Not included* | / | 2–4 servings | 1 |
| Sweets per week | <3 | 1 | ≤2 servings | 1 |
| Wine | ≥7 glasses per week | 1 | 1–2 glasses per day | 1 |
| Sweetened or carbonated beverages | <1 per day | 1 | *Included within sweets* | / |
| Butter, margarine, or cream per day | <1 serving | 1 | *Not included* | / |
| Sofrito sauce (made with tomato and onion, leek, or garlic and simmered with olive oil) | ≥2 per week | 1 | *Not included* | / |
| Potatoes per week | *Not included* | / | ≤3 servings | 1 |
| **Total score** | | **14** | | **24** |
| **Recommended cut-off points for determining MD adherence** | a) 3 groups: ≤5 points (low adherence), 6–9 points (moderate) and ≥10 points (high adherence) | | ≥13.5 refers to adherence | |
| | b) Binary: ≥8 denotes adherence | | | |

and nuts (2 points each), and 8 points for weekly intake of legumes, potatoes, eggs, fish, white meat, red meat and sweets (Table 1). According to the original study, people with a score of ≥13.5 on the MDSS scale can be considered as adherent to the principles of the MD, which we rounded up to 14 points (Table 1) [26]. The same MDSS scoring system was used in our previous study in the general population of Dalmatia, but the data were obtained through the FFQ [41]. We did not use the FFQ in this study due to the more extensive burden of this approach to the subjects. Instead, we have used only 14 questions including food groups that comprise the original MDSS index (S1 Table), with the exception that we didn't include beer in the fermented beverages, as originally proposed [26]. Instead, we only included wine, which is in accordance with the modern MD pyramid [16]. Additionally, the question on juices and sugar-sweetened beverages was introduced as a separate item, but it was scored within sweets, as proposed [26]. The full questionnaire is presented in both Croatian and English in S1 Table.

Mediterranean Diet Adherence Screener (MEDAS) was chosen as a gold standard needed to assess the validity of the MDSS index. Since MEDAS was not previously validated for application in Croatian language, we used the same steps for questionnaire preparation as for the MDSS questionnaire, and we also performed MEDAS test-retest reliability assessment.

The original version of MEDAS was designed and validated in Spain [23, 25]. It was translated into several languages and validated for use in Germany [28], Iran [29], UK [30], Turkey

[31], Korea [32] and Portugal [33]. The original version of the MEDAS questionnaire contains 14 items (S1 Table), with 12 questions about the frequency of food consumption, and two items are about the eating habits characteristic for the Spanish area [23]. Each item is scored with either a 0 or 1, with the overall score ranging between 0 and 14 (Table 1). There are two ways to categorize the overall MEDAS score. Subjects can be divided into 3 subgroups, where the score of ≤5 points indicates low adherence, 6–9 indicates moderate adherence and ≥10 points indicates high level of adherence to the principles of the MD (Table 1) [23, 25]. Additionally, a cut-off score of ≥8 points has been used to denote adherence to the principles of the MD, while MEDAS score of ≤7 points represents MD non-adherence [23]. Croatian version of the MEDAS questionnaire is presented in S1 Table.

There are some differences between MEDAS and MDSS questionnaires. For example, some food groups are included in MDSS and not in MEDAS, such as cereals, dairy products, eggs, and potatoes. MDSS separates fruit juices from fresh fruit consumption and it does not include processed meat, unlike MEDAS. On the other hand, MEDAS incorporates sofrito sauce, butter or margarine or cream and sweetened beverage intake as separate groups, whereas all types of juices are regarded as sweets according to the MDSS questionnaire scoring [26]. MEDAS distinguishes between cooked and raw vegetables and includes two questions on olive oil, which is not the case in MDSS index. Furthermore, there is a difference in proposed frequency of consumption for nuts, legumes, fish and red meat. MEDAS questionnaire aims to incorporate higher intake of traditional Mediterranean staples, such as vegetables, fruits, olive oil and fish, but it also takes into account some of the non-traditional, Western type of foods, such as already mentioned margarine or cream, processed meat and sweetened beverages, demanding their lower intake. On the other hand, MDSS questionnaire asks only about consumption of the traditional MD foods, entirely in accordance to the recommendations of the modern MD pyramid [16, 26]. MDSS is also giving more weight to the foods at the base of the MD pyramid and more points are awarded for higher intake of vegetables, fruit, cereals and olive oil, unlike in the MEDAS index. All of these differences between MDSS and MEDAS are presented in Table 1.

## Statistical analysis

Categorical variables were presented as absolute numbers and percentages. Numerical variables were mostly non-normally distributed (tested by Kolmogorov–Smirnov test), and they were presented as medians and interquartile ranges (IQR). Differences between groups were tested using chi-square test for categorical variables, and Mann–Whitney U test for numerical variables. Spearman rank test was used to test bivariate correlation between numerical variables.

Test-retest reliability was tested using intra-class correlation coefficients (ICC; two-way mixed model) and Spearman rank test for both MDSS and MEDAS overall scores. Based on the ICC estimates, values <0.50 were considered to show poor agreement, values between 0.50 and 0.75 as moderate, between 0.75 and 0.90 as good agreement, while values greater than 0.90 were regarded as excellent reliability [42]. Cohen's kappa statistic was used for assessing agreement between test-retest classification of subjects into tertiles and for MD adherence/non-adherence classification based on the appropriate cut-off points available in Table 1. According to McHugh et al., values ≤0 indicate that there is no agreement, and values 0.01–0.20 indicate that there is a slight agreement, 0.21–0.40 as fair agreement, 0.41–0.60 as moderate agreement, 0.61–0.80 as substantial agreement, and 0.81–1.00 as almost perfect agreement [43]. Additionally, test-retest agreement was calculated using kappa statistic for all of the separate food groups [43].

Concurrent validity of MDSS index was also tested using ICC, Spearman rank test and Cohen's kappa statistic, against MEDAS index, both for the first testing time, and for the retest. Despite methodological limitations, Spearman's rank test was calculated to provide comparison with previous studies.

We also applied the Principal Component Analysis (PCA) to both MD questionnaires to test construct validity and to identify food groups (factors), using Varimax rotation and the cut-off of >0.30 for absolute factor loadings to suppress small coefficients. The suitability of the data for factor analysis was tested by the Kaiser–Meyer–Olkin measure of sampling adequacy (≥0.60) and Bartlett's test of sphericity (P<0.05). Factors with an Eigenvalue ≥1.0 were retained, and total explained variance was recorded.

Due to missing data from the MD questionnaires, we excluded 17 medical students from the initial, exploratory sample (see Fig 2). This resulted in a sample size of 360 students at the first time point, included in concurrent validity analysis, while 210 of these students were available during the second time point (retest), and they were included in the test-retest reliability analysis. This was an appropriate sample size, based on the estimate of an ideal subject to questionnaire item ratio being between 10:1 and 20:1 [44].

All statistical analyses were performed using IBM SPSS Statistics for Windows, Version 21.0. (Armonk, NY: IBM Corp.). The level of significance was set at $P<0.05$.

## Ethical approval

The study was carried out following the Declaration of Helsinki and the protocol was approved by the Ethical Committee of the University Department of Health Studies (2181-228-07-19-0021) and the Ethical Committee of the University of Split School of Medicine (2181-198-03-04-18-0027).

## Results

### Sample characteristics

The exploratory sample of 360 medical students was included in the analysis, and 210 students were available for the retesting procedure (58.3%). The study included 248 women (71%) and 102 men (29%), while 10 students didn't provide information on the gender. There was no difference in MD compliance between men and women, assessed either with MEDAS ($P = 0.224$) or MDSS questionnaire ($P = 0.202$), even though women reported slightly higher MD adherence compared to men (21.4% vs 15.7%, respectively, for MEDAS questionnaire, and 14.9% vs 9.8% for MDSS questionnaire) (Table 2).

Students displayed different degree of adherence to MD food groups, ranging from 0.8% for wine in women to 92% for potatoes (both within MDSS index). Women reported higher compliance with olive oil intake, chicken, red/processed meat and sofrito sauce intake according to MEDAS questionnaire, and with red meat intake according to MDSS questionnaire, while men reported slightly higher adherence to wine intake, which was overall very low among students (Table 2). Less than half of the sample was adherent to the main staples of the MD, such as olive oil, cereals, vegetables, fruits and fish (Table 2).

### Repeatability (reliability) and validity analysis

Based on the ICC and correlation analysis, the test-retest reliability was very good for both MDSS questionnaire (ICC = 0.881, 95% CI 0.843–0.909, $P<0.001$; ρ = 0.627, $P<0.001$), and for MEDAS questionnaire (ICC = 0.887, 95% CI 0.852–0.914, $P<0.001$; ρ = 0.717, $P<0.001$) (Table 3).

**Table 2. Students' characteristics according to the gender (10 students didn't provide this information and they were excluded from this analysis).**

| | Men | Women | P |
|---|---|---|---|
| | N = 102 | N = 248 | |
| **Age (years); median (IQR)** | 19.0 (4.0) | 23.0 (5.0) | 0.200 |
| **Study year; N (%)** | | | |
| 1st | 46 (45.1) | 102 (41.1) | 0.341 |
| 3rd | 6 (5.9) | 18 (7.3) | |
| 5th | 50 (49.0) | 121 (48.8) | |
| Unknown | 0 (0.0) | 7 (2.8) | |
| **MEDAS test score (points); median (IQR)** | 5.0 (2.0) | 6.0 (3.0) | 0.039 |
| **MEDAS components; N (%)** | | | |
| Olive oil (yes) | 52 (51.0) | 160 (64.5) | 0.019 |
| Olive oil (tbsp.) | 17 (17.0) | 59 (24.1) | 0.150 |
| Vegetables | 48 (47.1) | 103 (41.5) | 0.343 |
| Fruits | 28 (27.7) | 56 (22.7) | 0.318 |
| Nuts | 38 (37.3) | 84 (33.9) | 0.546 |
| Legumes | 29 (28.4) | 76 (30.6) | 0.681 |
| Fish/shellfish | 6 (5.9) | 10 (4.0) | 0.451 |
| Chicken | 68 (66.7) | 198 (80.2) | 0.007 |
| Red/processed meat | 15 (14.7) | 78 (31.7) | 0.001 |
| Butter/margarine | 65 (63.7) | 163 (66.5) | 0.616 |
| Sweet beverages | 56 (54.9) | 148 (59.7) | 0.410 |
| Sweets | 42 (41.2) | 90 (36.3) | 0.391 |
| Wine | 5 (5.0) | 2 (0.8) | 0.013 |
| Sofrito sauce | 78 (76.5) | 217 (87.5) | 0.010 |
| **MEDAS adherence; N (%)** | | | |
| No (≤7 points) | 86 (84.3) | 195 (78.6) | 0.224 |
| Yes (≥8 points) | 16 (15.7) | 53 (21.4) | |
| **MEDAS adherence; N (%)** | | | |
| Low (≤5 points) | 57 (55.9) | 108 (43.5) | 0.096 |
| Moderate (6–9 points) | 41 (40.2) | 131 (52.8) | |
| High (≥10 points) | 4 (3.9) | 9 (3.6) | |
| **MDSS test score (points); median (IQR)** | 7.0 (6.0) | 8.0 (5.0) | 0.128 |
| **MDSS components; N (%)** | | | |
| Olive oil | 15 (15.0) | 52 (21.0) | 0.201 |
| Cereals | 47 (46.1) | 89 (36.6) | 0.090 |
| Vegetables | 18 (17.6) | 67 (27.0) | 0.063 |
| Fruits | 57 (55.9) | 161 (64.9) | 0.113 |
| Dairy products | 32 (31.4) | 77 (31.2) | 0.971 |
| Nuts | 13 (12.7) | 47 (19.0) | 0.162 |
| Legumes | 64 (62.7) | 174 (70.2) | 0.177 |
| Potatoes | 94 (92.2) | 229 (92.3) | 0.954 |
| Fish | 22 (21.6) | 53 (21.4) | 0.967 |
| Eggs | 63 (61.8) | 129 (52.4) | 0.111 |
| White meat | 15 (14.7) | 59 (23.9) | 0.056 |
| Red meat | 21 (20.6) | 95 (38.6) | 0.001 |
| Sweets | 33 (32.4) | 67 (27.3) | 0.348 |
| Wine | 6 (5.9) | 2 (0.8) | 0.004 |

*(Continued)*

**Table 2.** (Continued)

|  | Men | Women | P |
|---|---|---|---|
|  | N = 102 | N = 248 |  |
| **MDSS adherence; N (%)** |  |  |  |
| No | 92 (90.2) | 211 (85.1) | 0.202 |
| Yes | 10 (9.8) | 37 (14.9) |  |

In order to estimate the concurrent validity of the MDSS questionnaire expressed as an overall score, ICC was calculated with MEDAS score as a comparator. The agreement was moderate during both first testing session (ICC = 0.544, 95% CI 0.439–0.629, $P<0.001$), and the second testing session (ICC = 0.533, 95% CI 0.387–0.644; $P<0.001$) (Table 3).

As a comparator for reliability analysis, students' responses on the Warwick–Edinburgh Mental Well-being Scale (WEMWBS) were used. The correlation coefficient between the WEMWBS test and retest score was 0.800 ($P<0.001$), and ICC was 0.892 (95% CI 0.811–0.938, $P<0.001$). Performance of this questionnaire was tested as a binary variable, the sample was divided according to the median of 55 points into the subgroup below the median and the subgroup having the score equal or above the median. This yielded a kappa value of 0.581 ($P<0.001$) for test-retest repeatability of WEMWBS questionnaire.

Table 4 shows an estimate of concurrent validity of the MDSS index against the MEDAS index, when both were expressed as a binary variable. A fair agreement was demonstrated at both first time point ($\kappa = 0.205$; $P<0.001$) and the second assessment time point ($\kappa = 0.223$; $P<0.001$), while test-retest repeatability was better and showed a moderate agreement for MDSS ($\kappa = 0.584$; $P<0.001$), and substantial agreement for MEDAS questionnaire ($\kappa = 0.620$, $P<0.001$) (Table 4).

Subjects were further divided into distribution-defined tertiles according to both MEDAS and MDSS questionnaires scores obtained at the first and second assessment time point, and used in reliability and validity analysis. A fair agreement was demonstrated between MEDAS and MDSS tertile distribution only for the second assessment time point ($\kappa = 0.211$, $P<0.001$),

**Table 3.  Intra-class correlation coefficients (ICC) for test-retest reliability and concurrent validity of MDSS questionnaire expressed as an overall score (numerical variable).**

|  | MDSS test | MEDAS test | MEDAS retest |
|---|---|---|---|
| **MDSS retest** |  |  |  |
| ICC [95% C]; P | **0.881 [0.843–0.909]; <0.001** | 0.541 [0.398–0.650]; <0.001 | **0.533 [0.387–0.644]; <0.001** |
| Spearman rank rho (P) | 0.627 (<0.001) | 0.488 (<0.001) | 0.490 (<0.001) |
| **MEDAS retest** |  |  |  |
| ICC [95% C]; P | 0.507 [0.353–0.625]; <0.001 | **0.887 [0.852–0.914]; <0.001** | n/a |
| Spearman rank rho (P) | 0.408 (<0.001) | 0.717 (<0.001) |  |
| **MDSS test** |  |  |  |
| ICC [95% C]; P | n/a | **0.544 [0.439–0.629]; <0.001** | n/a |
| Spearman rank rho (P) |  | 0.391 (0.004) |  |

ICC–intra-class correlation coefficient, n/a–not applicable

**Table 4. Test-retest reliability and concurrent validity of the MDSS questionnaire expressed as the binary variable (originally proposed cut-off values were applied for both MEDAS and MDSS).**

| | MDSS test | | MEDAS test | | MEDAS retest | |
|---|---|---|---|---|---|---|
| | N (%) | | N (%) | | N (%) | |
| **MDSS retest** | **No** | **Yes** | **No** | **Yes** | **No** | **Yes** |
| No; N (%) | 168 (80.0) | 10 (4.8) | 146 (69.5) | 32 (15.2) | 150 (71.4) | 28 (13.3) |
| Yes; N (%) | 12 (5.7) | 20 (9.5) | 19 (9.1) | 13 (6.2) | 19 (9.1) | 13 (6.2) |
| κ (P) | **0.584 (<0.001)** | | 0.194 (0.004) | | **0.223 (<0.001)** | |
| **MEDAS retest** | No | Yes | No | Yes | No | Yes |
| No; N (%) | 152 (72.4) | 17 (8.1) | 154 (73.3) | 15 (7.2) | | |
| Yes; N (%) | 28 (13.3) | 13 (6.2) | 11 (5.2) | 30 (14.3) | | |
| κ (P) | 0.241 (<0.001) | | **0.620 (<0.001)** | | n/a | |
| **MDSS test** | No | Yes | No | Yes | No | Yes |
| No; N (%) | | | 260 (72.2) | 51 (14.2) | | |
| Yes; N (%) | | | 29 (8.0) | 20 (5.6) | | |
| κ (P) | n/a | | **0.205 (<0.001)** | | n/a | |

while test-retest repeatability showed a moderate agreement for both MDSS (κ = 0.447, $P<0.001$) and MEDAS questionnaire (κ = 0.511, $P<0.001$) (S2 Table).

Table 5 shows the agreement between adherence to all of the food groups represented within MDSS and MEDAS questionnaires, based on the appropriate cut-off points (see Table 1). Kappa values for agreement between MDSS and MEDAS questionnaires varied considerably across food groups, with substantial agreement obtained only for sweets during the retesting, moderate agreement for red meat and wine, fair agreement for legumes and fish, while for other food groups we found none or only slight agreement (Table 5).

**Table 5. Agreement between adherence to the food groups comprising the MD, assessed by MDSS and MEDAS questionnaires; data are presented as κ values (P values) for each of the food group included in the questionnaires and expressed as binary variables (adherent/non-adherent).**

| | MDSS vs. MEDAS test agreement | MDSS vs. MEDAS retest agreement | MEDAS test vs. MEDAS retest agreement | MDSS test vs. MDSS retest agreement |
|---|---|---|---|---|
| | κ (P value) | κ (P value) | κ (P value) | κ (P value) |
| **Olive oil (y/n)** | 0.096 (<0.001) | 0.095 (<0.001) | 0.782 (<0.001) | - |
| **Olive oil (tbsp)** | 0.142 (<0.001) | 0.156 (<0.001) | 0.753 (<0.001) | 0.724 (<0.001) |
| **Cereals** | - | - | - | 0.596 (<0.001) |
| **Vegetables** | 0.126 (<0.001) | 0.130 (<0.001) | 0.566 (<0.001) | 0.694 (<0.001) |
| **Fruits** | 0.118 (<0.001) | 0.083 (<0.001) | 0.622 (<0.001) | 0.672 (<0.001) |
| **Nuts** | 0.164 (<0.001) | 0.185 (<0.001) | 0.572 (<0.001) | 0.775 (<0.001) |
| **Legumes** | 0.293 (<0.001) | 0.261 (<0.001) | 0.493 (<0.001) | 0.612 (<0.001) |
| **Fish** | 0.242 (<0.001) | 0.365 (<0.001) | 0.566 (<0.001) | 0.746 (<0.001) |
| **White meat** | -0.056 (0.051) | -0.055 (0.132) | 0.830 (<0.001) | 0.397 (<0.001) |
| **Red meat** | 0.405 (<0.001) | 0.399 (<0.001) | 0.648 (<0.001) | 0.647 (<0.001) |
| **Sweets** | 0.606 (<0.001) | 0.756 (<0.001) | 0.590 (<0.001) | 0.816 (<0.001) |
| **Wine** | 0.387 (<0.001) | 0.489 (<0.001) | 1.00 (<0.001) | 0.793 (<0.001) |
| **Dairy** | - | - | - | 0.654 (<0.001) |
| **Eggs** | - | - | - | 0.686 (<0.001) |
| **Potatoes** | - | - | - | 0.641 (<0.001) |
| **Butter/margarine** | - | - | 0.668 (<0.001) | - |
| **Sweetened beverages** | - | - | 0.784 (<0.001) | - |
| **Sofrito sauce** | - | - | 0.545 (<0.001) | - |

Test-retest agreement for food groups within each of the questionnaire was moderate or even better. Furthermore, MDSS questionnaire performed comparably or even better than MEDAS questionnaire for most of the food groups, except for the white meat agreement ($\kappa$ = 0.397 for MDSS vs. $\kappa$ = 0.830 for MEDAS questionnaire) (Table 5).

Finally, in order to assess construct validity of both MD indexes, factor analysis was performed, for which the appropriateness of the data was supported by the Kaiser–Meyer–Olkin measure of sampling adequacy (0.60 for MEDAS, and 0.61 for MDSS) and Bartlett's test of sphericity ($P<0.001$, both MEDAS and MDSS). Six food factors (each with Eigenvalue >1) were extracted for both MEDAS and MDSS questionnaire, using principal component analysis (Table 6). This accounted for 59.3% of the total variance in the Mediterranean dietary pattern assessed by the MEDAS questionnaire, and 60.2% of the total variance assessed by the MDSS questionnaire. Identified factors and their corresponding loading values showed substantial overlap between MEDAS and MDSS index. For example, factor 1 included vegetables and fruit for both MD indexes, factor 2 included olive oil and fish, while factor 3 contained both read and white meat, along with some other foods (Table 6).

## Predictive validity

To assess predictive validity and to confirm the initial agreement between MEDAS and MDSS indexes, we included a confirmatory sample of 299 students attending health studies. MEDAS questionnaire yielded slightly higher prevalence of the MD adherence (14.7%) compared to MDSS index (9.4%) (Table 7).

There was no difference according to the gender, while subjects who were adherent to the principles of the MD were on average slightly older. There was no difference between adherent and non-adherent subjects in their BMI, the same as for the number of meals per day, breakfast frequency and smoking habits (Table 7). MD adherent subjects according to MDSS index were more frequently preparing their own food ($P = 0.003$), and 69.6% of them have been adhering to a weight loss diet at some point in their life compared to 46.9% in non-adherent subjects ($P = 0.037$), but without such differences for MEDAS index. No differences were found for snacking habits during TV watching and for body appearance satisfaction for either of the MD index. On the other hand, there was a difference in sports and gym-using habits, where subjects adherent to MD according to MEDAS index were more frequently physically active compared to non-adherent subjects (Table 7).

**Table 6. Identified factors and their corresponding loading values for MEDAS and MDSS indexes (principal component analysis).**

|  | MEDAS index food groups (factor loadings) | MDSS index food groups (factor loadings) |
|---|---|---|
| **Factor 1** | Vegetables total (0.867), raw vegetables (0.866), fruit (0.415) | Fruit (0.718), vegetables (0.682), legumes (0.676), nuts (0.581) |
| **Factor 2** | Olive oil for cooking (0.796), olive oil quantity in tablespoons (0.782), fish/shellfish (0.558) | Fish (0.816), olive oil (0.696) |
| **Factor 3** | Sweet beverages (0.663), red meat/processed meat (0.583), sweets/pastries (0.509), white meat instead of red meat (-0.497), butter (0.425) | Red meat (0.750), potatoes (0.732), white meat (0.490), wine (0.454) |
| **Factor 4** | Nuts (0.684), fruit (0.462), butter (0.451), white meat instead of red meat (0.390), legumes (0.382), fish/shellfish (0.360) | Eggs (0.733), dairy (0.707), white meat (0.333) |
| **Factor 5** | Sofrito sauce (0.755), legumes (0.525) | Cereals (0.794), white meat (0,340), wine (-0.324), dairy (-0.315) |
| **Factor 6** | Wine (0.756), butter (0.459), sweets (-0.390) | sweets (0.770), wine (-0.613) |

**Table 7. Lifestyle characteristics associated with the Mediterranean diet assessed by MEDAS and MDSS questionnaires (predictive validity) in the confirmatory sample of health studies students.**

| | MEDAS questionnaire | | P | MDSS questionnaire | | P |
|---|---|---|---|---|---|---|
| | Non-adherent | Adherent | | Non-adherent | Adherent | |
| | N = 255 (85.3%) | N = 44 (14.7%) | | N = 271 (90.6%) | N = 28 (9.4%) | |
| **Sex; N (%)** | | | | | | |
| Men | 36 (14.1) | 7 (15.9) | 0.754 | 40 (14.8) | 3 (10.7) | 0.561 |
| Women | 219 (85.9) | 37 (84.1) | | 231 (85.2) | 25 (89.3) | |
| **Age; median (IQR)** | 20.0 (3.0) | 22.0 (14.0) | 0.005 | 21.0 (6.0) | 22.0 (14.0) | 0.012 |
| **BMI; median (IQR)** | 22.2 (3.7) | 21.7 (3.8) | 0.495 | 22.0 (3.7) | 22.3 (4.3) | 0.235 |
| **Weighing (days ago); median (IQR)** | 15.0 (26.0) | 10.0 (28.0) | 0.216 | 15.0 (26.0) | 10.0 (27.0) | 0.456 |
| **Breakfast (days/week); median (IQR)** | 5.2 (3.0) | 7.0 (4.4) | 0.949 | 6.0 (3.0) | 7.0 (4.0) | 0.953 |
| **Meals per day; median (IQR)** | 4.3 (1.7) | 4.0 (1.6) | 0.546 | 4.3 (1.7) | 4.3 (1.7) | 0.767 |
| **Smoking; N (%)** | | | | | | |
| Yes | 68 (28.3) | 11 (27.5) | 0.193 | 73 (28.5) | 6 (25.0) | 0.231 |
| Ex-smokers | 29 (12.1) | 9 (22.5) | | 32 (12.5) | 6 (25.0) | |
| Never smoked | 143 (59.6) | 20 (50.0) | | 151 (59.0) | 12 (50.0) | |
| **Cooking; N (%)** | | | | | | |
| Yes, frequently | 67 (27.9) | 16 (41.0) | 0.213 | 70 (27.3) | 13 (56.5) | 0.003 |
| Sometimes | 133 (55.4) | 19 (48.7) | | 147 (57.4) | 5 (21.7) | |
| No | 40 (16.7) | 4 (10.3) | | 39 (15.2) | 5 (21.7) | |
| **Weight loss diet; N (%)** | | | | | | |
| Yes | 113 (47.1) | 23 (59.0) | 0.168 | 120 (46.9) | 16 (69.6) | 0.037 |
| No | 127 (52.9) | 16 (41.0) | | 136 (53.1) | 7 (30.4) | |
| **Snacking while watching TV; N (%)** | | | | | | |
| Yes, frequently | 53 (22.1) | 6 (15.4) | 0.175 | 54 (21.1) | 5 (21.7) | 0.725 |
| Yes, sometimes | 143 (59.6) | 21 (53.8) | | 152 (59.4) | 12 (52.2) | |
| No | 44 (18.3) | 12 (30.8) | | 50 (19.5) | 6 (26.1) | |
| **Satisfied with body appearance; N (%)** | | | | | | |
| Yes | 128 (53.3) | 21 (53.8) | 0.943 | 139 (54.3) | 10 (43.5) | 0.529 |
| No | 70 (29.2) | 12 (30.8) | | 73 (28.5) | 9 (39.1) | |
| Didn't think about it | 42 (17.5) | 6 (15.4) | | 44 (17.2) | 4 (17.4) | |
| **Sports; N (%)** | | | | | | |
| Yes, weekly | 68 (34.5) | 19 (54.3) | 0.026 | 78 (36.4) | 9 (50.0) | 0.254 |
| Rarely or never | 129 (65.5) | 16 (45.7) | | 136 (63.6) | 9 (50.0) | |
| **Gym; N (%)** | | | | | | |
| Yes, weekly | 31 (15.7) | 12 (34.3) | 0.009 | 39 (18.2) | 4 (22.2) | 0.675 |
| Rarely or never | 166 (84.3) | 23 (65.7) | | 175 (81.8) | 14 (77.8) | |
| **Self-assessed health perception; median (IQR)** | 8.0 (1.0) | 8.5 (2.0) | 0.435 | 8.0 (2.0) | 9.0 (2.0) | 0.050 |
| **Quality of life; median (IQR)** | 8.0 (2.0) | 7.5 (3.0) | 0.348 | 8.0 (1.0) | 8.0 (2.0) | 0.062 |
| **Mental well-being (WEMWBS); median (IQR)** | 52.0 (11.0) | 54.0 (10.0) | 0.833 | 53.0 (10.0) | 56.0 (17.0) | 0.048 |

There was a borderline insignificant difference in self-assessed health perception, which was slightly higher in subjects who followed the principles of the MD as measured by MDSS index ($P = 0.050$), with similar result for the mental well-being score ($P = 0.048$) (Table 7).

To confirm the validity of the MDSS questionnaire in health studies subjects (confirmatory sample), the agreement analysis yielded a moderate agreement when using the overall score (ICC = 0.510, 95% CI 0.384–0.610; $P<0.001$), while kappa value for adherence agreement was 0.216 ($P<0.001$). Furthermore, a statistically significant correlation was found between the

**Table 8. Correlation between MEDAS and MDSS scores and lifestyle factors, perception of health and quality of life (data are Spearman rank rho coefficients and _P_ values).**

| | MDSS score | Age | BMI | Weighing (days ago) | Breakfast (days/week) | Meals per day | Self-assessed health perception | Quality of life | WEMWBS |
|---|---|---|---|---|---|---|---|---|---|
| **MEDAS score** | 0.486 | 0.021 | 0.046 | -0.083 | 0.081 | -0.051 | 0.116 | 0.060 | 0.094 |
| | <**0.001** | 0.730 | 0.460 | 0.179 | 0.187 | 0.413 | 0.063 | 0.326 | 0.122 |
| **MDSS score** | - | 0.179 | -0.001 | -0.076 | 0.093 | -0.015 | 0.123 | 0.109 | 0.139 |
| | | **0.003** | 0.981 | 0.217 | 0.130 | 0.805 | **0.047** | 0.073 | **0.022** |
| **Age** | | - | 0.192 | -0.139 | -0.229 | -0.265 | 0.007 | 0.076 | 0.173 |
| | | | **0.002** | **0.024** | <**0.001** | <**0.001** | 0.907 | 0.210 | **0.004** |
| **BMI** | | | - | -0.098 | -0.147 | -0.191 | -0.056 | -0.102 | 0.001 |
| | | | | 0.114 | **0.017** | **0.002** | 0.370 | 0.099 | 0.994 |
| **Weighing (days ago)** | | | | - | -0.028 | 0.057 | -0.143 | -0.115 | -0.105 |
| | | | | | 0.649 | 0.361 | **0.023** | 0.062 | 0.088 |
| **Breakfast (days/week)** | | | | | - | 0.464 | 0.064 | 0.131 | -0.015 |
| | | | | | | <**0.001** | 0.308 | **0.032** | 0.812 |
| **Meals per day** | | | | | | - | 0.080 | 0.086 | -0.012 |
| | | | | | | | 0.206 | 0.163 | 0.844 |
| **Self-assessed health perception** | | | | | | | - | 0.479 | 0.285 |
| | | | | | | | | <**0.001** | <**0.001** |
| **Quality of life** | | | | | | | | | 0.476 |
| | | | | | | | | | <**0.001** |

MEDAS score and the MDSS score (ρ = 0.486: _P_<0.001) (Table 8). Additionally, MDSS score was positively correlated with age (ρ = 0.179: _P_ = 0.003), self-assessed health perception (ρ = 0.123; _P_ = 0.047), and mental well-being score (WEMWBS) (ρ = 0.139; _P_ = 0.022), while this was absent for the MEDAS score (Table 8). Self-assessed health perception was positively correlated with the quality of life (ρ = 0.479; _P_<0.001), while mental well-being score was correlated with both health perception (ρ = 0.285; _P_<0.001) and the quality of life (ρ = 0.476; _P_<0.001) (Table 8).

## Discussion

We showed a very good reliability of the overall MDSS score, while the reliability of the MD adherence as a binary variable was moderate. Validity of the MDSS index, compared to the MEDAS index as a referent point, was also better when expressed as a total score than adherence. These results were replicated in our confirmatory sample, verifying our findings that the MDSS questionnaire is a reasonably valid instrument for the MD assessment in Croatia.

Differing results obtained for the overall numerical scores and for the binary adherence are in line with previous papers, which suggested that dichotomizing a continuous variable is not the optimal way of handling the data [45, 46]. Indeed, we have shown here that the use of a continuous variable provides much better estimates and should be favored as opposed to categorization.

To the best of our knowledge, this is the first study to compare the MDSS with the MEDAS questionnaire for validation purposes. We have chosen the MEDAS questionnaire as a referent point due to several reasons. Firstly, it includes similar questions and food groups and employs a similar scoring approach to the MDSS questionnaire. Furthermore, it was already shown to be valid in several languages [28–33], and it is quite frequently used in the literature. However, there are some differences between those two indexes. These differences have affected our

results, yielding lower agreements and validity estimates within specific food groups. For example, based on the comparison of MDSS questionnaire items with MEDAS items, only sweets, red meat and wine had moderate or better agreement, legumes and fish had fair agreement, while the remaining five food groups had none to only slight agreement. In light of previously mentioned differences between MDSS and MEDAS questionnaire, this was not a surprising finding. However, factor analysis revealed that both questionnaires explained similar proportion of the total variance of the MD pattern, 59.3% for MEDAS and 60.2% for MDSS questionnaire. Furthermore, identified factors and their corresponding loading values showed remarkable overlap between the two indexes. This confirms that both questionnaires are assessing the universal Mediterranean dietary pattern.

Given that we used the MEDAS questionnaire as a referent point, we had to prepare this questionnaire to be used in Croatian language, and we used the same procedures as we did for the MDSS questionnaire. Additionally, we tested the performance of MEDAS questionnaire regarding its reliability. Our results showed that MEDAS questionnaire displayed a close to excellent repeatability in the test-retest procedure, with intra-class correlation coefficient of 0.887, which was very similar to the result obtained for MDSS. This high reliability can be compared to the MEDAS validation study carried out in the UK, where two administrations of the MEDAS produced similar mean total scores and an intra-class correlation coefficient of 0.69 [30]. When we analyzed MEDAS item-by-item test-retest agreement, we found the highest agreement coefficients for wine and white meat, followed by sweetened beverages and olive oil, while all other questions/components showed appropriate, moderate agreement. MDSS questionnaire showed higher test-retest agreement for vegetables, fruits, nuts, legumes, fish, and sweets than the same items in the MEDAS questionnaire, while a slightly lower agreement was recorded only for wine and olive oil and a notably lesser agreement for white meat. Hence, we can conclude that MDSS questionnaire items performed comparably or even better than MEDAS questionnaire items in test-retest repeatability for most of the food groups, except for the white meat.

Brief dietary questionnaires are useful and commonly used for identifying the overall eating patterns as well as for highlighting problems with patient's eating habits easily and quickly. Unfortunately, there are so many dietary questionnaires in use, particularly those for assessing the MD pattern. For example, a recent study identified as many as 28 different indexes being used in the literature, but only a very few scores fulfilled the applicability parameters and psychometric quality, while the overall level of evidence was scarce [34]. Other reviews have also found similar abundance of indexes used in the literature, and performed comparison of performance of several commonly used dietary indexes and questionnaires [8, 20, 47–52]. These studies have shown that there is no such thing as one ideal instrument, which would objectively measure the adherence to the MD [50]. This represents a certain barrier in both research and practical domains, disabling direct comparisons between studies and across populations, as well as contributing to the lack of indisputable evidence of MD benefits, which could be used for MD endorsement on the larger population scale. For, example, a recent Cochrane systematic review on the MD use for the primary and secondary prevention of cardiovascular diseases pointed out the variety of MD definitions, low to moderate quality of evidence and only modest benefits of the MD [53].

In previous validation studies, MEDAS questionnaire tended to yield a higher MD adherence than the control questionnaires [25, 28, 30]. This was also the case in our study. MEDAS showed a slightly higher rate of MD adherence compared to the MDSS in both of our samples, but overall it was still quite low. According to the MEDAS questionnaire, 19.7% of medical students were compliant with the MD, and as few as 13.6% according to the MDSS questionnaire. In the confirmatory sample of health studies students we recorded 14.7% of MD adherent

students based on the MEDAS questionnaire and 9.4% based on MDSS index. This is lower than our expectations, especially when we take into account that our samples consisted of the future healthcare professionals, who are expected to be educated in the matters of disease prevention and to have the prominent role in terms of care and education of patients about healthy lifestyle and health protection. Despite their biomedical educational background, these young people reported similar, low prevalence of MD adherence just as the general young population of Dalmatia [41]. Unfortunately, our results are not an isolate finding [54]. The same pattern of decay of MD lifestyle and diet, being replaced by the Western lifestyle, was found among university students in other countries [55–63]. A recent study found that students who live away from their parents and those from Mediterranean countries deviate from MD towards more unhealthy diet, while future healthcare professionals were neither familiar with the Mediterranean diet nor were following the principles of the MD [64]. On the other hand, the same study found that lower MD adherence in students was associated with poorer health status, while higher MD adherence was associated with lower depression risk and better sleep quality [64]. Our results have confirmed these findings, since we found that students with higher MDSS score had better self-assessed health perception and mental well-being. These findings alone could and should be used in health education and MD promotion in student population, which is clearly needed.

One of the limitations of this study is a cross-sectional design for the part of the study investigating predictive validity of the MD questionnaire. Furthermore, data collection was carried out in such a way that subjects were required to recall their eating habits, which could have resulted in the recall bias. Most importantly, we did not use face to face interviews like previous validation studies did [25, 33], but instead we used a self-administered questionnaire. We believed that our anonymous and self-administering approach would enhance the response rate and facilitate honest responses from students, while not substantially diminishing credibility and reliability of the data (a facilitator was always present and students could inquire about any uncertainties).

It is also important to note that our sample included a younger population comprised exclusively of students, who are mostly healthy, while previous validation studies were largely carried out on a sample of older adults at risk for various chronic diseases [25, 28, 30].

The strengths of the study include a strict methodological framework, two relatively large and independent samples with high response rate (≥80%) from Dalmatia county, which were assumed to have more uniform and traditional eating habits. This is the first study to compare the MDSS with MEDAS questionnaire for validation purposes and the first validation study of the Croatian version of the short MD questionnaire for adult population. The recommendations state that the questionnaires should be linguistically adapted to the country in which they are used in order to ensure reliability and acceptable level of validity of the obtained results [24, 34]. Furthermore, the rationale for adapting different instruments to measure MD adherence in different populations and the comparison of how and why these instruments perform differently are interesting and important questions to explore and warrant further studies. This study described performance of two short MD screeners and it answers to an unmet need of Croatian scientists and clinicians, providing a valid and easy to use instrument for assessing MD adherence. The MDSS questionnaire could be used as a screening tool in the general population for public health surveillance, in clinical settings, and in scientific studies.

## Conclusions

We demonstrated that a short version of the MDSS questionnaire is highly reliable and a reasonably valid instrument for the assessment of the adherence to the overall Mediterranean

dietary pattern in Croatia. The best performance of MDSS was obtained when used as a numeric score, even in the population with low MD adherence. Despite above-mentioned limitations, the Croatian version of the short 14-items MDSS questionnaire can be used for a rapid assessment of adherence to the MD in Croatia, possibly both in research and in clinical practice.

## Supporting information

**S1 Checklist.**
(DOCX)

**S1 Table. Croatian and English versions of the MDSS [26] and MEDAS [23] questionnaires.**
(DOCX)

**S2 Table. Agreement between classification of the participants into tertiles according to the MEDAS and MDSS questionnaire scores.**
(DOCX)

## Acknowledgments

We would like to thank students of the University of Split (University Department of Health Studies, and School of Medicine) for their participation in this study. We are thankful to Marina Lukezic, MD for language editing.

## Author Contributions

**Conceptualization:** Ozren Polašek, Ivana Kolčić.

**Data curation:** Mario Marendić, Ivana Kolčić.

**Formal analysis:** Ozren Polašek, Ivana Kolčić.

**Investigation:** Mario Marendić, Nikolina Polić, Helena Matek, Lucija Oršulić, Ivana Kolčić.

**Methodology:** Mario Marendić, Ozren Polašek, Ivana Kolčić.

**Project administration:** Mario Marendić, Ivana Kolčić.

**Supervision:** Ivana Kolčić.

**Visualization:** Ivana Kolčić.

**Writing – original draft:** Mario Marendić, Ivana Kolčić.

**Writing – review & editing:** Mario Marendić, Nikolina Polić, Helena Matek, Lucija Oršulić, Ozren Polašek, Ivana Kolčić.

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
