## [Decision Letter · Decision Letter 0]

30 Nov 2020

PONE-D-20-26357

Mediterranean Diet Assessment Challenges: Validation of the Croatian Version of the 14-item Mediterranean Diet Serving Score (MDSS) Questionnaire

PLOS ONE

Dear Dr. Kolčić,

Thank you for submitting your manuscript to PLOS ONE. After careful consideration, we feel that it has merit but does not fully meet PLOS ONE’s publication criteria as it currently stands. Therefore, we invite you to submit a revised version of the manuscript that addresses the points raised during the review process.

We look forward to receiving your revised manuscript.

Kind regards,

Cristina Vassalle

Academic Editor

PLOS ONE

Journal Requirements:

5. Please include captions for your Supporting Information files at the end of your manuscript, and update any in-text citations to match accordingly. Please see our Supporting Information guidelines for more information: http://journals.plos.org/plosone/s/supporting-information

Reviewers' comments:

Reviewer's Responses to Questions

**Comments to the Author**

1. Is the manuscript technically sound, and do the data support the conclusions?

Reviewer #1: Yes

Reviewer #2: No

Reviewer #3: Partly

2. Has the statistical analysis been performed appropriately and rigorously? 

Reviewer #1: Yes

Reviewer #2: N/A

Reviewer #3: No

3. Have the authors made all data underlying the findings in their manuscript fully available?

Reviewer #1: Yes

Reviewer #2: No

Reviewer #3: Yes

4. Is the manuscript presented in an intelligible fashion and written in standard English?

Reviewer #1: Yes

Reviewer #2: No

Reviewer #3: Yes

5. Review Comments to the Author

Reviewer #1: The article presents a very pertinent topic and is very well written. However, some changes are necessary in relation to the body of the article:

* the introduction of the article is very well based, but it is too long. As this is a validation article, it is of utmost importance that the methods and results are well explored (as they are in the article), but this makes the article long. My suggestion is to reduce the size of the introduction, so that reading the article does not become tiring.

* In the methods, explain better what are the main discrepancies between the translated versions.

* Figures 1 and 2 are of very low quality, making viewing difficult.

* There are errors of references in the text that must be corrected (lines 228, 231, 235, 239 and 246)

Reviewer #2: Review of PONE-D-20-26357: Mediterranean Diet Assessment Challenges: Validation of the Croatian Version of the 14-item Mediterranean Diet Serving Score (MDSS) Questionnaire

The authors propose to validate a short Mediterranean diet screener (MDSS) against another Mediterranean diet screener (MEDAS). They provide a comprehensive, up-to-date review of the literature on the Mediterranean diet, its association with health outcomes, and highlight the need for tools to efficiently measure it.

I do not, however, agree with the premise that one MD dietary screener can be used to validate another. There is a lack of concordance between the original purposes of the screeners, the questions/items included in the screeners, and the criteria for adherence on the items. These are two quite different instruments and ways of defining and measuring Mediterranean diet adherence, as is discussed in detail in lines 437-459. MEDAS was developed in Spain and used to test adherence to the intervention in the PREDIMED trial; where in 1 arm participants were asked to consume 4 TBSP of olive oil per day – and provided with the oil – which represents a higher intake of olive oil than typically found in other studies of the Spanish population. The PREDIMED researchers themselves modified the screener when they used it in a weight loss trial. As such, it does not seem reasonable to consider it (as is) the ‘gold standard’ for measuring MD adherence in all populations, despite the fact that it has been used and ‘validated’ against comprehensive dietary assessment measures in several different populations. The MDSS, based on the latest update of the MD pyramid, might be a better candidate for a generalizable screener, although might still need adaptation for specific populations/countries (as the paper title suggests was done for this study). The question of the rationale for adapting or developing different instruments to measure MD adherence in different populations/countries, and the comparison how and why these instruments perform differently is an interesting and important question to explore; and would be a more appropriate use of this data set.

As a minor note, the English in the paper was generally good, but did have a few grammatical and spelling errors; so the authors' work would benefit from professional English editing, particularly when submitted to a journal like PLOS ONE that does not copyedit final manuscripts.

Reviewer #3: This is a cross-sectional study involving two independent samples, of University students, in order to assess psychometric properties of the Croatian version of a short Mediterranean Diet questionnaire. The issue is of interest giving the importance of investigating eating habits on a large scale with simple but reliable tools. The work is well designed and the reading is sliding, however there are some methodological concerns and questions.

Minor issue:

1) The authors cited the paper of D.B. Panagiotakos where it is explained: “We used 11 main components of the Mediterranean diet (non-refined cereals, fruits, vegetables, potatoes, legumes, olive oil, fish, red meat, poultry, full fat dairy products and alcohol” [https://doi.org/10.1016/j.numecd.2005.08.006].

The MDSS investigate the consumption of dairy product without distinguish low from medium/high content of fat (i.e. yogurt from cheese). This could results in a leak of goodness in assessing adherence to MD. Did the authors consider these occurrence in choosing MDDS? Should they explain how they can assert the adherence to MD without accounting for this distinction?

2) Similar to the above. Still referring to D.B. Panagiotakos, the MDDS des not distinguish whole from refined grains. Also if refined grains are largely consumed among population, whole grains use is the reference for the adherence to MD. Should the authors have chosen another questionnaire for their validation paper?

Major issue:

1) The authors tested the reliability of MEDAS and MDSS performing the test-retest procedure, but they didn’t assess the internal consistency of the MDDS (i.e. estimated with Cronbach’s alpha). Could the author provide this part in the methodology section?

2) Could the author, also, provide the psychometric properties of the items? (inter-item association)

6. PLOS authors have the option to publish the peer review history of their article (what does this mean?). If published, this will include your full peer review and any attached files.

Reviewer #1: No

Reviewer #2: No

Reviewer #3: No

---

## [Author Response · Author response to Decision Letter 0]

17 Jan 2021

Response to the Editor: 

Answer: We appreciate your comment; we have checked the PLOS ONE's style requirements again, to ensure these have been followed out. We provide:

1. A marked-up copy of our manuscript with highlighted changes made to the original version ('Revised Manuscript with Track Changes_PONE-D-20-26357').

2. An unmarked version of our revised paper without tracked changes ('Manuscript_PONE-D-20-26357').

Answer: We used this phrase for highlighting that those results were not included in the tables, since they would not suffice as a standalone table, but the entire result was reported in the text. We have deleted these phrases (two of them, page 17, line 362 & page 22, line 435; in the manuscript with track changes), based on your comment. Additionally, we have uploaded our data to the Figshare repository (DOI 10.6084/m9.figshare.13560497, which was also cited in the Manuscript within ‘Supporting information’ on page 40.)

Answer: We’ve done as stated earlier, and uploaded the data to the Figshare repository (DOI: 10.6084/m9.figshare.13560497).

4. PLOS requires an ORCID iD for the corresponding author in Editorial Manager on papers submitted after December 6th, 2016. Please ensure that you have an ORCID iD and that it is validated in Editorial Manager. To do this, go to ‘Update my Information’ (in the upper left-hand corner of the main menu), and click on the Fetch/Validate link next to the ORCID field. This will take you to the ORCID site and allow you to create a new iD or authenticate a pre-existing iD in Editorial Manager. Please see the following video for instructions on linking an ORCID iD to your Editorial Manager account:

https://www.youtube.com/watch?v=_xcclfuvtxQ

Answer: We provided ORCID ID of the corresponding author during re-uploading documents in Editorial Manager (https://orcid.org/0000-0001-7918-6052). 

5. Please include captions for your Supporting Information files at the end of your manuscript, and update any in-text citations to match accordingly. Please see our Supporting Information guidelines for more information: http://journals.plos.org/plosone/s/supporting-information

Answer: Thank you for drawing our attention to this, we’ve corrected existing errors regarding the captions for ‘Table 1’, ‘Table 2, as well as for ‘Supplemental Table 1 and 2’. We have also moved the section ‘Supporting information’ to the end of manuscript, and included captions for our Supporting Information, it can be found under lines 849-853. 

 

Response to Reviewer 1: 

The article presents a very pertinent topic and is very well written. However, some changes are necessary in relation to the body of the article:

1. The introduction of the article is very well based, but it is too long. As this is a validation article, it is of utmost importance that the methods and results are well explored (as they are in the article), but this makes the article long. My suggestion is to reduce the size of the introduction, so that reading the article does not become tiring.

Answer: We have done our best to shorten the Introduction, from the initial 1050 words to 820 words now. We hope it is now acceptable.

2. In the methods, explain better what are the main discrepancies between the translated versions.

Answer: Previously we have included this explanation in the ‘Discussion’ section. We have now moved the text from ‘Discussion’ section, and we now describe the differences between the questionnaires in the Methods in subsection ‘MD assessment instruments’ (lines 262 to 278).

3. Figures 1 and 2 are of very low quality, making viewing difficult.

• Answer: We appreciate your comment, we tried to correct this. We uploaded our figures files (1 and 2) to the Preflight Analysis and Conversion Engine (PACE) digital tool, https://pacev2.apexcovantage.com/, to ensure that our figures meet PLOS requirements. PACE Adjustments for uploaded Figure 1 was: dimensions adjusted to 7.5in W x 3.79in H; DOCX file was converted to a valid TIF file, and we have done the same procedure for Figure 2. PACE Adjustments for uploaded Figure 2 was: dimensions adjusted to 7.35in W x 8.74in H; DOCX file was converted to a valid TIF file. We hope that visibility will be better now.

4. There are errors of references in the text that must be corrected (lines 228, 231, 235, 239 and 246)

Answer: Thank you for this observation. There were some mistakes with the file that was originally submitted. We’ve corrected existing errors in the captions for ‘Table 1’, ‘Table 2’ and for ‘Supplemental Table 1’, which can be found in the text within the lines 236, 237, 241, 246, 253, 261, 342 on page 11, 12 and 16. We’ve also corrected existing error in the caption for ‘Supplemental table 2’, which can be found in lines 246 and 380 of the revised manuscript with track changes.

 

Response to Reviewer 2: 

The authors propose to validate a short Mediterranean diet screener (MDSS) against another Mediterranean diet screener (MEDAS). They provide a comprehensive, up-to-date review of the literature on the Mediterranean diet, its association with health outcomes, and highlight the need for tools to efficiently measure it.

1. I do not, however, agree with the premise that one MD dietary screener can be used to validate another. There is a lack of concordance between the original purposes of the screeners, the questions/items included in the screeners, and the criteria for adherence on the items. These are two quite different instruments and ways of defining and measuring Mediterranean diet adherence, as is discussed in detail in lines 437-459. MEDAS was developed in Spain and used to test adherence to the intervention in the PREDIMED trial; where in 1 arm participants were asked to consume 4 TBSP of olive oil per day – and provided with the oil – which represents a higher intake of olive oil than typically found in other studies of the Spanish population. The PREDIMED researchers themselves modified the screener when they used it in a weight loss trial. As such, it does not seem reasonable to consider it (as is) the ‘gold standard’ for measuring MD adherence in all populations, despite the fact that it has been used and ‘validated’ against comprehensive dietary assessment measures in several different populations. The MDSS, based on the latest update of the MD pyramid, might be a better candidate for a generalizable screener, although might still need adaptation for specific populations/countries (as the paper title suggests was done for this study). The question of the rationale for adapting or developing different instruments to measure MD adherence in different populations/countries, and the comparison how and why these instruments perform differently is an interesting and important question to explore; and would be a more appropriate use of this data set.

Answer: Thank you for your comment, we think it is of substantial importance for our study. We are aware of the challenges of the Mediterranean diet assessment, and this is one of the main reasons we conducted this study, along with the fact that we still do not have a MD questionnaire for adults, which was validated in Croatian language. 

We have carefully considered many possible options and methodological approaches during study design phase, and unfortunately, none of them was a perfect solution. Given that our target population included students, we could not have encumbered them with too long and demanding questionnaires because we would not have reached large enough response rate and sample size. That was the main reason why we chose MEDAS questionnaire as a comparator, since we wanted to do more than just a repeatability analysis for the MDSS questionnaire. Indeed, MEDAS was originally developed for the purposes of the PREDIMED study, but it is now used as one of the common screeners of MD adherence, and it was shown to be a valid instrument, as you also pointed out. To accentuate and substantiate this statement, we added this section in the Introduction: “This questionnaire transcended its original use in the Spanish population, and it has been used widely in various cultural and societal settings. Several validation studies showed that MEDAS is a valid and reliable questionnaire in different countries and languages [33-38].“ (lines 105-107). It is also worth mentioning that both MEDAS and MDSS questionnaires were developed in Spain. 

In your comment, you mention the issue of consumption of olive oil according to MEDAS criterion, which you state to be representing a higher intake of olive oil than typically found (4 TBSP of olive oil per day). Our results in olive oil intake according to MEDAS and MDSS are very similar, and even showed a higher compliance with olive oil intake according to the MEDAS, compared to MDSS criterion. According to MEDAS criterion, we recorded 17% of men and 24% of women who reported taking at least 4 TBSP of olive oil per day, while according to MDSS criterion we showed that 15% of men and 21% of women were adherent, i.e. taking olive oil several times a day.

Despite differences between MDSS and MEDAS questionnaires in the specific questions, as described in details and now included into Methods section (lines 262-278), we have shown that both questionnaires are measuring the same, universal Mediterranean dietary pattern. Based on the result of the factor analysis, both questionnaires explained similar proportion of the total variance of the MD pattern (59.3% for MEDAS and 60.2% for MDSS questionnaire), and both questionnaires yielded the same factors with similar corresponding loading values. This indicates a remarkable overlap between the two indexes and we can conclude that they are in essence not so different. This explanation can be found in the Discussion section (lines 484-488). 

 In addition, even if MDSS was assessed as not in ideal alignment with MEDAS in our concurrent validity analysis (ICC=0.544 [0.439-0.629]); κ=0.223), we have still obtained satisfactory result, and especially so for the reliability and predictive validity of the MDSS index. Even though you are completely right when stating that MDSS “might be a better candidate for a generalizable screener” for MD, and we agree with this entirely, we still needed a comparison with gold standard to be able to assess the performance of MDSS. And even though we got ‘less than perfect’ results in concurrent validity analysis, we consider this as a lesser issue in general, especially when compared to the possibility of the overestimation of validity (hypothetical). Additionally, thank you for pointing this out, we have used your comment in our manuscript since we think it is very important idea. We have added the sentence: " Furthermore, the rationale for adapting different instruments to measure MD adherence in different populations and the comparison of how and why these instruments perform differently are interesting and important questions to explore and warrant further studies.” lines 559-562). Indeed, we have assessed the performance of both MEDAS and MDSS instrument in our population (we also provide MEDAS test-retest reliability). 

Furthermore, this is actually the first study aiming to validate a short and easy to apply MD questionnaire, and the first one to our knowledge to be used in adults in Croatia (only KIDMED was validated previously, and only test–retest reliability was assessed; https://www.mdpi.com/2072-6643/9/4/419). We know that MD assessment instrument is truly needed, and we have already received several inquiries from our colleagues about a validated questionnaire they could use in their studies. Moreover, we plan to use this questionnaire in our future papers and projects, given that it is in line with the MD pyramid and that it is easy to administer, and was shown to be associated with various health outcomes (as we have found in some of our previous papers: https://doi.org/10.1108/BFJ-06-2018-0339; doi: 10.3390/nu9121296.; doi: 10.3390/nu12041164.; doi: 10.3390/nu13010097.). 

In conclusion, we hope that our methodological approach can be acceptable, regardless of the limitations, given that we have included several validity parameters in the same study (which is rarely the case in the literature), and that we have employed two independent and relatively large samples. In this approach, we have performed confirmatory analysis of our results, finding them to be stable. This study answers to an unmet need of Croatian scientists and clinicians, providing a valid and easy to use instrument for assessing MD adherence (the last sentence was added to the Discussion to point this out; lines 562-564).

2. As a minor note, the English in the paper was generally good, but did have a few grammatical and spelling errors; so the authors' work would benefit from professional English editing, particularly when submitted to a journal like PLOS ONE that does not copyedit final manuscripts

Answer: A native speaker has checked the language, and we have included her in the ‘Acknowledgments’ section on page 29.

 

Response to Reviewer 3: 

This is a cross-sectional study involving two independent samples, of University students, in order to assess psychometric properties of the Croatian version of a short Mediterranean Diet questionnaire. The issue is of interest giving the importance of investigating eating habits on a large scale with simple but reliable tools. The work is well designed and the reading is sliding, however there are some methodological concerns and questions.

1. The authors cited the paper of D.B. Panagiotakos where it is explained: “We used 11 main components of the Mediterranean diet (non-refined cereals, fruits, vegetables, potatoes, legumes, olive oil, fish, red meat, poultry, full fat dairy products and alcohol” [https://doi.org/10.1016/j.numecd.2005.08.006].

Answer: We appreciate your comment, but we didn’t originally cite this paper in our manuscript. However, we have included it now in Introduction and Discussion, since it is relevant for our topic. Thank you for bringing this paper to our attention.

2. The MDSS investigate the consumption of dairy product without distinguish low from medium/high content of fat (i.e. yogurt from cheese). This could results in a leak of goodness in assessing adherence to MD. Did the authors consider these occurrence in choosing MDDS? Should they explain how they can assert the adherence to MD without accounting for this distinction?

Answer: Thank you for this comment, we do agree with your premise about the need for distinguishing low from medium/high content of fat in dairy products. However, the questionnaire that was used in a study by authors Panagiotakos DB et al., asks subjects about “full fat dairy products”, while MDSS asks about “Milk and dairy products”, including milk, yoghurt, cheese, and even ice-cream. This is due to the fact that the modern MD pyramid, which was the basis for MDSS score development is putting more emphasis on low fat dairy products (Figure 2 in paper by Bach-Faig A, Berry EM, Lairon D, Reguant J, Trichopoulou A, Dernini S, et al. Mediterranean diet pyramid today. Science and cultural updates. Public Health Nutr. 2011;14(12A):2274-84. doi: https://doi.org/10.1017/S1368980011002515). While you may in fact have the point in advantageous use of full fat dairy, which is based on ‘Traditional Mediterranean diet pyramid, and as mentioned and incorporated into the Mediterranean diet score proposed by Panagiotakos DB et al, we could not change the original MDSS index too much (we only excluded beer consumption), and we decided to keep it as similar to the originally proposed scoring system and the underlying MD pyramid.

Furthermore, there are some differences in food consumption between Croatia and Greece. Greeks mostly use medium/high-fat dairy products daily i.e. feta cheese, while for Croatian people this is not so typical. In conclusion, we kept the scoring system within the originally proposed frame and we hope this can be acceptable, given that this is in line with the MD pyramid (Bach-Faig A, et al. doi: https://doi.org/10.1017/S1368980011002515). 

3. Similar to the above. Still referring to D.B. Panagiotakos, the MDDS des not distinguish whole from refined grains. Also if refined grains are largely consumed among population, whole grains use is the reference for the adherence to MD. Should the authors have chosen another questionnaire for their validation paper?

Answer: Yes, MDSS questionnaire asks about consumption of ‘Cereals, all types (bread, breakfast cereals, pasta and rice)’. MDSS index does not distinguish between whole grain and non-whole grain cereals, probably due to the fact that MD pyramid recommends whole grains, but does not demand them (Bach-Faig A et al). Based on this, we didn’t ask our subjects about consumption of cereals separately in this study. 

Even though this is an important matter, we have still decided to use MDSS index as the one that would be applicable in our population. There are several reasons for that. The first one is that it is in line with the modern MD pyramid (Bach-Faig A et al). The second reason is that it covers the main components of the MD, some of which are not even included in MEDAS questionnaire (for example cereals, dairy, eggs and potatoes). Thirdly, MDSS puts greater emphasis on the most important food groups, those that form the basis of MD pyramid, and which should be included daily (more points are awarded to those food groups). Finally, it is short and very easy to apply and score (for both respondents and for those who apply it), making it quick and practical, while providing a general overview of the MD pattern. Hence, it could be used as a screening tool in the general population for public health surveillance, in clinical settings with patients, and in scientific studies.

4. The authors tested the reliability of MEDAS and MDSS performing the test-retest procedure, but they didn’t assess the internal consistency of the MDDS (i.e. estimated with Cronbach’s alpha). Could the author provide this part in the methodology section?

Answer: This is a fair point, given that most of the papers dealing with questionnaires report this parameter. There are several reasons why we didn’t initially perform this analysis. Firstly, and of minor significance, Cronbach’s alpha was not reported in the original paper describing and validating MDSS questionnaire (Monteagudo C, Mariscal-Arcas M, Rivas A, Lorenzo-Tovar ML, Tur JA, Olea-Serrano F. Proposal of a Mediterranean Diet Serving Score. PLoS ONE. 2015;10(6):e0128594.). Secondly, both MDSS and MEDAS questionnaires are comprised of 14 questions/food groups, and all of these questions are recoded into binary variables (adherent /not adherent to a particular food group intake), which would demand another kind of test for internal consistency (Kuder-Richardson 20 test), while Cronbach’s alpha is more appropriate for Likert scales or ordinal variables. Finally, and most importantly, we have revealed six factors underlying the MDSS questionnaire in our sample (Table 6), and we got the same result for MEDAS questionnaire. Thus, performing Cronbach’s alpha analysis in our study would break the first premise of the Cronbach’s alpha parameter, which is the unidimensionality of the instrument. In a paper by Taber KS on the use of Cronbach’s alpha it was stated that “the scale needs to be unidimensional to provide an “interpretable” result, as a score obtained from a measuring scale ought to indicate the “amount” of the construct being measured” (Taber KS. The Use of Cronbach’s Alpha When Developing and Reporting Research Instruments in Science Education. Res Sci Educ. 2018;48;1273–1296. https://doi.org/10.1007/s11165-016-9602-2). Furthermore, in that paper the author concludes: “Cronbach’s alpha is most valuable for indicating scale reliability in the sense of the equivalence of items within single-construct scales, but the statistic does not offer any indication that scales are actually unidimensional (which should be tested by other means).” (Taber KS).

 However, we have performed this analysis, merely as an experiment and out of curiosity and we did get a result that is supporting our finding that the MDSS instrument applied in our sample was indeed multi-dimensional, since we got the Cronbach's alpha values of 0.249 for MDSS questionnaire, and it was 0.403 for MEDAS questionnaire. This is indeed in line with the statement from a recent paper by McNeish D: “In many circumstances, violating these (unrealistic) assumptions yields estimates of reliability that are too small, making measures look less reliable than they actually are.” (McNeish D. Thanks coefficient alpha, we'll take it from here. Psychol Methods. 2018;23:412-433. doi: 10.1037/met0000144.). Additionally, Sijtsma K. also stated that “probably no other statistic has been reported more often as a quality indicator of test scores than Cronbach’s alpha coefficient, and presumably no other statistic has been subject to so much misunderstanding and confusion (Sijtsma K. On the Use, the Misuse, and the Very Limited Usefulness of Cronbach's Alpha. Psychometrika. 2009;74(1):107-120. doi:10.1007/s11336-008-9101-0). There are some studies, which conclude that Cronbach’s alpha is commonly used erroneously (Gardner PL. Measuring attitudes to science: unidimensionality and internal consistency revisited. Research in Science Education. 1995;25:283–289. doi:10.1007/bf02357402.), and sometimes it is abused and contributes to the confusion in the literature (Sijtsma K.; doi:10.1007/s11336-008-9101-0), and we would like to avoid contributing to this situation further by reporting Cronbach’s alpha. These are the reasons why we would like to ask to be exempted from reporting Cronbach’s alpha in our study. Furthermore, instead of Cronbach’s alpha we reported three types of questionnaire reliability and validity parameters: 1. test-retest reliability (both intra-class correlation coefficients and Cohen’s kappa values were calculated), 2. concurrent validity, and 3. predictive validity. Proving the unidimensionality of the MDSS or MEDAS index was not our aim and we consider this less relevant for the purpose of our study.

5. Could the author, also, provide the psychometric properties of the items? (inter-item association)

Answer: This question is the continuation of the previous one, and hence the answer we provided in the previous sections is also applicable here. However, we have performed this analysis (using binary variables), and here is the table for MDSS index, indicating that some of the variables have a negative correlation and work in the opposite direction, instead of contributing equally to the underlying construct. This again supports the argument that our results are departing from the unidimensionality of the MDSS questionnaire. 

Inter-Item Correlation Matrix

 Mdss_fruit mdss_veggies mdss_cereals mdss_potato mdss_oliveoil mdss_nuts mdss_dairy mdss_legumes mdss_eggs mdss_fish mdss_whitemeat mdss_redmeat mdss_sweets mdss_wine

mdssfruit 1.000 .194 .008 -.017 -.064 .226 .028 .254 .010 .099 .080 .109 -.008 -.056

mdss_veggies .194 1.000 .032 -.030 -.026 .086 .041 .225 -.027 .097 -.105 .059 .120 -.007

mdss_cereals .008 .032 1.000 -.089 .067 -.060 .145 -.059 .036 -.043 .101 -.193 -.131 .129

mdss_potato -.017 -.030 -.089 1.000 -.185 .050 -.006 -.069 -.094 .052 -.076 .116 .068 -.086

mdss_oliveoil -.064 -.026 .067 -.185 1.000 -.022 -.035 -.072 -.056 -.024 -.027 -.036 -.031 .332

mdss_nuts .226 .086 -.060 .050 -.022 1.000 .123 .128 .040 .028 .013 .102 .056 -.073

mdss_dairy .028 .041 .145 -.006 -.035 .123 1.000 .018 .098 -.018 -.029 .089 -.010 -.069

Mdss_legumes .254 .225 -.059 -.069 -.072 .128 .018 1.000 .115 .005 .029 .084 .002 .035

mdss_eggs .010 -.027 .036 -.094 -.056 .040 .098 .115 1.000 .012 .033 -.081 .008 .001

mdss_fish .099 .097 -.043 .052 -.024 .028 -.018 .005 .012 1.000 .005 -.011 .133 .089

mdss_whitemeat .080 -.105 .101 -.076 -.027 .013 -.029 .029 .033 .005 1.000 -.050 .003 .003

mdss_redmeat .109 .059 -.193 .116 -.036 .102 .089 .084 -.081 -.011 -.050 1.000 .109 -.037

mdss_sweets -.008 .120 -.131 .068 -.031 .056 -.010 .002 .008 .133 .003 .109 1.000 .097

mdss_wine -.056 -.007 .129 -.086 .332 -.073 -.069 .035 .001 .089 .003 -.037 .097 1.000

 Additional changes we made in the manuscript: Language editing

---

## [Decision Letter · Decision Letter 1]

4 Feb 2021

Mediterranean Diet Assessment Challenges: Validation of the Croatian Version of the 14-item Mediterranean Diet Serving Score (MDSS) Questionnaire

PONE-D-20-26357R1

Dear Dr. Kolčić,

We’re pleased to inform you that your manuscript has been judged scientifically suitable for publication and will be formally accepted for publication once it meets all outstanding technical requirements.

Kind regards,

Cristina Vassalle

Academic Editor

PLOS ONE

Additional Editor Comments (optional):

Reviewers' comments:

Reviewer's Responses to Questions

**Comments to the Author**

1. If the authors have adequately addressed your comments raised in a previous round of review and you feel that this manuscript is now acceptable for publication, you may indicate that here to bypass the “Comments to the Author” section, enter your conflict of interest statement in the “Confidential to Editor” section, and submit your "Accept" recommendation.

Reviewer #1: All comments have been addressed

Reviewer #3: All comments have been addressed

2. Is the manuscript technically sound, and do the data support the conclusions?

Reviewer #1: Yes

Reviewer #3: Yes

3. Has the statistical analysis been performed appropriately and rigorously? 

Reviewer #1: Yes

Reviewer #3: Yes

4. Have the authors made all data underlying the findings in their manuscript fully available?

Reviewer #1: Yes

Reviewer #3: Yes

5. Is the manuscript presented in an intelligible fashion and written in standard English?

Reviewer #1: Yes

Reviewer #3: Yes

6. Review Comments to the Author

Reviewer #1: (No Response)

Reviewer #3: I really appreciate that the authors performed a test to assess the internal consistency of the MDSS, and I thanks them, even though the Cronbach’s alpha was mentioned as an example “(i.e. estimated with Cronbach’s alpha)” and did not necessarily have to be adopted given the limitation of this instrument in a multidimensional scale, as the authors reported.

Despite the original paper of Monteagudo C established a cut-off, I guess if in your population, as the authors stated, it would be more effective to use the questionnaire score as continuous rather than dichotomized. In fact the authors show that the score loads for 6 factors (tab 6) therefore it is not properly correct to use the dichotomous score, but this is just a personal reflection.

Your paper is well designed and described, it is suitable for the publication in PLOS ONE.

7. PLOS authors have the option to publish the peer review history of their article (what does this mean?). If published, this will include your full peer review and any attached files.

Reviewer #1: No

Reviewer #3: No

---

## [Editor Report · Acceptance letter]

8 Feb 2021

PONE-D-20-26357R1 

Mediterranean Diet Assessment Challenges: Validation of the Croatian Version of the 14-item Mediterranean Diet Serving Score (MDSS) Questionnaire 

Dear Dr. Kolčić:

I'm pleased to inform you that your manuscript has been deemed suitable for publication in PLOS ONE. Congratulations! Your manuscript is now with our production department. 

Kind regards, 

on behalf of

Dr. Cristina Vassalle 

Academic Editor

PLOS ONE